# Contribution of Proteins to the Latin American Diet: Results of the ELANS Study

**DOI:** 10.3390/nu15030669

**Published:** 2023-01-28

**Authors:** Marianella Herrera-Cuenca, Martha Cecilia Yépez García, Lilia Yadira Cortés Sanabria, Pablo Hernández, Yaritza Sifontes, Guillermo Ramírez, Maura Vásquez, Georgina Gómez, María Reyna Liria-Domínguez, Attilio Rigotti, Mauro Fisberg, Irina Kovaslkys, Maritza Landaeta-Jiménez

**Affiliations:** 1Centro de Estudios del Desarrollo, Universidad Central de Venezuela (CENDES-UCV), Caracas 1053, Venezuela; 2Fundación Bengoa, Caracas 1053, Venezuela; 3Colegio de Ciencias de la Salud, Universidad San Francisco de Quito, Quito 17-1200-841, Ecuador; 4Departamento de Nutrición y Bioquímica, Pontificia Universidad Javeriana, Bogotá 110231, Colombia; 5Escuela de Nutrición y Dietética, Facultad de Medicina, Universidad Central de Venezuela, Caracas 1053, Venezuela; 6Área de Postgrado en Estadística, Facultad de Ciencias Económicas y Sociales, Universidad Central de Venezuela, Caracas 1053, Venezuela; 7Departamento de Bioquímica, Escuela de Medicina, Universidad de Costa Rica, San José 11501-2060, Costa Rica; 8Instituto de Investigación Nutricional, La Molina, Lima 15026, Peru; 9Centro de Nutrición Molecular y Enfermedades Crónicas, Departamento de Nutrición, Diabetes y Metabolismo, Escuela de Medicina, Pontificia Universidad Católica, Santiago 8330024, Chile; 10Centro de Excelencia em Nutrição e Dificuldades Alimentaes (CENDA), Instituto Pensi, Fundação José Luiz Egydio Setubal, Hospital Infantil Sabará, São Paulo 01228-200, Brazil; 11Departamento de Pediatria, Universidade Federal de São Paulo, São Paulo 04023-061, Brazil; 12Carrera de Nutrición, Facultad de Ciencias Médicas, Pontificia Universidad Católica Argentina, Buenos Aires C1107 AAZ, Argentina

**Keywords:** diet, Latin America, protein intake, animal protein, vegetable protein, processed protein, essential amino acids, ELANS

## Abstract

Dietary protein intake is vital to life. Here we sought to characterize dietary sources of protein in eight Latin American countries. Survey data were collected for Estudio Latinoamericano de Nutrición y Salud (ELANS); participants were from Argentina, Brazil, Chile, Colombia, Costa Rica, Ecuador, Peru, and Venezuela (*n* = 9218, 15–65 years old). The primary aim of this analysis was to quantify per-person daily protein consumption by country and sociodemographic factors. Secondary aims: to quantify proportional intake of proteins by source, amount and processing, and to determine the adequacy of protein/essential amino acid intake. Younger groups (adolescents 15–19 years, adults 20–33 years) had the highest intake of proteins; middle-aged adults (34–49 years) had a lower intake, and older adults (50–65 years) had a strikingly lower intake. Protein consumption was higher in men than women. Animal proteins comprised nearly 70% of total daily protein intake in Argentina and Venezuela, contrasting with <60% in Peru, Chile, and Costa Rica. Brazil and Venezuela showed the highest protein intake within the highest education level. The higher the socioeconomic level, the higher the protein intake, except for Argentina, Chile, and Peru. Proportional intake of animal- and plant-based protein generally reflected the food availability by country. This study presents a pre-pandemic regional baseline and offers a perspective for future studies of changes related to government policies, climate, and dietary practices.

## 1. Introduction

Dietary protein intake is vital to life, and adequate intake is essential to growth, health, and functionality. Consumed proteins are hydrolyzed into small peptides and amino acids in the lumen of the gastrointestinal tract [1]. Such digestion products are absorbed into enterocyte cells lining the intestine, then further transported to the liver and other tissues for resynthesis as new proteins for cell structure and function. In addition, the metabolism of the amino acid glutamine yields energy as adenosine triphosphate (ATP) for lymphocytes and macrophages, thus sustaining immune function [2]. Amino acids are also metabolized to form essential non-protein cellular components and regulators, e.g., purines, pyrimidines, and neurotransmitters [3]. Protein undernutrition in adults can thus lead to compromised health conditions such as anemia, physical weakness, poor wound healing, vascular dysfunction, and impaired immunity [4].

Global and national health leaders, including the World Health Organization (WHO), have developed guidelines for the quantity and quality of protein intake across the human lifespan [3,4,5,6,7]. For most adults, recommended intake is 0.8 g/kg body weight/day, although a higher level of 1.0–1.2 g/kg/day is advised for healthy people older than 65 years or even higher (1.2–1.5 g/kg/day) for those who have acute or chronic illnesses [6,8,9,10]. A good quality protein is one with an amino acid composition covering the RDA requirement for each individual essential amino acid [7]. Factors that affect protein quality include the source (animal versus plant) and preparation for intake (processing and cooking) [3,7]. While the evidence is strong regarding the role of dietary protein in maintaining health across the lifespan, specific qualities of proteins from different dietary sources are not yet fully understood or agreed upon [2,11,12]. Plant proteins, when compared with animal proteins, have recently attracted considerable attention in terms of healthiness, affordability, and production sustainability [11,12,13,14,15,16]. It is thus important to understand patterns of protein intake (amounts, sources, and processing) in populations of Latin America.

In addition, without going in-depth on the ongoing discussion of whether plant-based proteins might be more friendly to the environment, it should be mentioned that it is an important issue regarding the problems that challenge humanity’s well-being. Recent research assessing what the associated factors for climate change are has developed toward the gas emissions produced by cattle farming, pork farming, and even fish farming and the various effects that those might have on the environment [17]. However, mono-agriculture extensions have arisen as a competitor to animal farming, and the recent geopolitical conflicts have taken a toll on the planet’s health as well [18].

Currently, the controversy on whether plant-based proteins might be more friendly to the environment needs to be discussed in extension because, on the one hand, the production of animal protein products represents a potential increase in greenhouse gas emissions, while on the other, extensive mono-agriculture shows the challenges associated with adding greenhouse emissions, soil damage, and both plants and animals, elevate the waste [19]. All these compromise the sustainability of food supply chains, with the particular concern of whether there will be enough protein to guarantee humanity’s well-being [17]. In addition, the recent geopolitical conflicts, such as the Russia-Ukraine war, among other protracted armed conflicts less covered by the media at the moment, such as Syria, Afghanistan, and South Sudan, have taken a toll on the planet’s health as well, with a massive disruption on the global food market by increasing the prices of goods, the interruption of the commercialization chains and local environmental damage. Therefore shortages of foods in general, and particularly for those inhabitants of such regions, are expected, including the protein supplies [18].

To the best of our knowledge, there are few studies that integrate regional food consumption data in Latin America following the same methodology. In addition, the fact that the world is experiencing a pandemic of COVID-19 justifies the analysis of these data to be used as a pre-pandemic regional baseline.

The aims of this analysis were to (i) quantify per-person daily protein consumption by country and by sociodemographic factors and to establish between countries variability; (ii) to quantify proportional intake of proteins by food groupings, protein source, and processing; and (iii) to determine the adequacy of protein/essential amino acid intake. We used dietary protein data collected in a survey of participants from Argentina, Brazil, Chile, Colombia, Costa Rica, Ecuador, Peru, and Venezuela (*n* = 9218, 15–65 years old). These data provide a baseline perspective that can be used as a comparator for future studies of diet and health-related conditions to government policies, climate, and dietary practices.

## 2. Materials and Methods

### 2.1. Survey Methods

The Latin American Study of Nutrition and Health (Estudio Latinoamericano de Nutrición y Salud, ELANS) was conducted as a household-based, multi-national, cross-sectional survey over a period of one year in eight Latin American countries (i.e., Argentina, Brazil, Chile, Colombia, Costa Rica, Ecuador, Peru, and Venezuela). All study sites were academic institutions in urban areas of the countries. Researchers followed a common study protocol for interviewer training, implementation of fieldwork, data collection and management, and quality control procedures, as detailed elsewhere [20,21].

### 2.2. Data Collection

All data were collected between October 2014 and 2015. Data were stratified by country, sex, age, education level, and socioeconomic level to build a representative sample of urban household populations in the studied countries. For educational level, the categories of basic education only, secondary school degree, and university graduate were used. Socioeconomic levels were identified as high, middle, and low using scales appropriate to each country.

A procedure based on short-term reporting, 24-h dietary recalls, was used to estimate usual dietary intake according to the Multiple Pass Method to guarantee that no step of the recall was forgotten and ensure quality control [22]. The 24-h recall was selected for its general applicability and relatively easy format to be responded to by interviewees. To guarantee the consistency of the food intake recall, 2 non-consecutive 24-h recalls were performed. The trained interviewers attained detailed information on all food and beverages, preparations, and supplements consumed [20]. Portions of consumed food and beverages were transformed with the software Nutrition Data System for Research (NDS-R, Minnesota University, Minneapolis, MN, USA. Version 2013) into macro and micronutrients. In correspondence with the study’s objectives, we addressed only protein and essential amino acid intake. The web-based statistical modeling technique Multiple Source Method, proposed by the European Prospective Investigation into Cancer and Nutrition (EPIC) [23], was used to estimate usual protein (g) and essential amino acid (mg) intake.

Dietary protein was divided into animal protein and vegetable protein (including grains, legumes, nuts, and other nonanimal sources); and processed and unprocessed meat. Processed meat was defined as meat treated through salting, curing, fermentation, smoking, or other processes to enhance flavor and improve preservation, e.g., ham, bacon, smoked beef, or pork.

To examine specific consumption patterns, 9 protein-containing food groups were recognized: (1) dairy, (2) eggs, (3) beef, (4) poultry, (5) fish, (6) pork, (7) cereals, (8) legumes, and (9) nuts and seeds.

### 2.3. Data Analysis and Statistics

Descriptive statistics were computed for continuous measures as means, standard deviations, 95% confidence intervals (CI), and categorical measures as counts and percentages. Data were analyzed by fitting a general linear model with protein intake as the dependent variable; sociodemographic factors and second-order interactions were used as explanatory variables. The calculation procedure for amino acid adequacy is as follows: the observed value was divided by the reference and multiplied by 100. If this number was between 90 and 110, the subject was classified as normal; if <90, the subject’s amino acid intake was inappropriate due to deficit; and if >110, the value was inappropriate due to excess. A linear discriminant analysis was performed to obtain groups of countries according to the consumption of processed and unprocessed meats.

To investigate the similarity of essential amino acid intake between countries, principal component analysis (PCA) [24,25] was performed based on essential amino acid quantities consumed.

Descriptive statistical analyses and the general linear model fitting were performed using the statistical program SPSS Statistics for Windows v25 (SPSS v25, IBM Corporation, Armonk, NY, USA).

### 2.4. Ethical Issues

The complete ELANS protocol was registered at Clinical Trials (#NCT02226627) and was approved by the Western Institutional Review Board (#20140605). Site-specific protocols were further approved by the ethical review boards of participating institutions. Participants provided informed consent for inclusion in the country-level studies, and participant confidentiality was maintained by the use of numeric identification codes rather than names. Data transfers were conducted by way of a secure file-sharing system.

## 3. Results

### 3.1. Overview

Protein consumption was characterized by (i) country of residence and sex, (ii) sociodemographic variables including educational level and economic level, (iii) protein sources as specific food groups, animal- versus plant-based protein, processed versus unprocessed protein, and by (iv) dietary sufficiency of essential amino acids.

### 3.2. Average Daily Protein Consumption According to Sex and Country

Between-country differences in the mean protein intake (*p*-value < 0.001) led to a classification of participating countries into three groups (Figure 1). Ecuador and Argentina had the highest daily protein intake in the region, with a range of 85.5–87.5 g/day as 95% confidence limits. The populations of Colombia, Peru, Brazil, and Venezuela had similar mid-range protein intakes of 78.6–79.8 g/day, while Costa Rica and Chile had the lowest daily protein intake range of 66.8–68.8 g/day.

The protein intake, as described by the interaction of sex and country, confirmed that average protein consumption was higher in men than in women. However, sex-related differences in intake varied significantly by country in the region (*p*-value < 0.001). In Argentina and Brazil, men consumed between 15 g/day and 20 g/day more protein than women. In Chile, Costa Rica, Ecuador, and Peru, the values for protein consumption by men were between 10 g/day and 18 g/day higher than in women. In Colombia and Venezuela, the gaps in protein intake between the two sexes were smaller; men exceeded women for intake amounts by only 8 g/day to 13 g/day.

### 3.3. Sociodemographic Factors That Can Affect Protein Intake: Age, Educational Level, and Socio-Economic Level

By age, estimated protein intake was highest in younger groups (15 to 19 years old and 20 to 34 years), then tended to decrease with age in midlife (34 to 49 years) and decrease markedly among older adults (50 to 65 years), thus yielding a pattern of significant decline with age (*p*-value < 0.001; Figure 2). This pattern of protein consumption was seen in Venezuela, but other countries had protein intake profiles that differed statistically from the overall population pattern. In Argentina, there were no significant differences in protein consumption by age group. In Brazil, Colombia, Peru, and Chile, the group of older adults (50 to 65 years) had significantly lower protein intake in comparison with the other three younger age groups, while in Ecuador and Costa Rica, the youngest group of adults consumed significantly more protein than people in all three older groups.

By educational level (basic education only, secondary school degree, university graduate) and by country, protein intake was not widely differentiated. Only two countries showed notable differences. In Brazil, the population group with a basic educational level had a significantly lower protein intake compared to groups with higher educational levels. Similarly, adults in Venezuela with the highest level of education had significantly higher protein consumption than those of the other two educational groups.

In general, the average protein intake increases progressively with the ascent in the social scale, finding higher levels of protein consumption in the middle and upper socioeconomic levels in some countries. The interaction of socioeconomic level (SEL) with the country indicates that: (i) in Ecuador, Colombia, Brazil, Venezuela, and Costa Rica, the upper and middle SEL had a higher protein intake than the lower SEL; (ii) in Chile, the low SEL had a slightly higher protein intake than the other two SEL; and (iii) in Argentina and Peru there were no differences in protein consumption across SEL (Figure 3).

### 3.4. Estimated Proportional Intake of Plant- and Animal-Based Protein

Expressed within 95% CI, adults in Peru have the highest average level for daily adult intake of plant-based protein (31.4–32.4 g/day), followed by those living in Costa Rica, Ecuador, and Chile. Argentina and Colombia had an even lower average daily intake of plant-based protein (25.7–26.7 g/day; 26.1–26.8 g/day, respectively), and consumption in Brazil was lower still (24.6–25.3 g/day). Of all the countries studied, adults in Venezuela had the lowest daily intake of plant-based protein (95% CI: 23.0–23.9 g/day). By comparison, Argentina is the country with the highest average level for adult daily intake of animal-based protein (58.7–60.7 g/day), followed by Ecuador, Venezuela, Brazil, and Colombia. Peru has the next lowest level (45.3–47.0 g/day), and Costa Rica and Chile have the lowest levels (37.7–39.9 g/day; 38.1–39.8 g/day). See online Appendix A.

Proportional intake of plant- and animal-based dietary protein are shown in the next figure (Figure 4). In Argentina and Venezuela, animal protein was nearly 70% of total daily protein intake, while in Brazil, Colombia, and Ecuador, intake was between 66 and 68%. By contrast, animal protein was less than 60% in Peru, Chile, and Costa Rica.

Detailed tables about protein intake by sociodemographic factors, protein sources, and country are included as supplementary tables alongside this publication (See online Appendix A).

### 3.5. Between-Country Comparison of Protein Consumption by Food Group

Results are shown for proteins in the six food groups that were consumed most, i.e., poultry, beef, fish, pork, dairy, and eggs (Figure 5). Animal-based foods with the greatest contribution to protein intake were beef and poultry. Argentina and Brazil stand out in this regard, with a consumption of 29 g/day of beef and approximately 20 g/day of poultry. They are followed by Colombia, Ecuador, and Venezuela, with 20 to 24 g/day of poultry and a lower intake of beef between 20 to 22 g/day. Chile showed a lower level of intake of poultry (14 g/day). Peru differs from the rest of the countries, with a high consumption of poultry (28 g/day) and only 7 g/day of beef. As for dairy, pork, eggs, and fish, protein intake is less than 15 g/day in all countries. Ecuador (12 g/day) and Peru (11 g/day) stand out, with the highest consumption of fish and Peru with the lowest consumption of pork (4 g/day).

In all countries, the plant-based protein food groups with the highest contribution to protein consumption were grains (above 15 g/day) (Figure 6). In Argentina, Chile, and Peru, this consumption exceeds 20 g/day. It is followed by the consumption of legumes, which varies between 1 g/day (Argentina) and 10 g/day (Costa Rica). Nuts and seeds provide a protein consumption of less than 1 g/day.

### 3.6. Protein Intake from Processed and Unprocessed Meats

A linear discriminant analysis was applied to establish similarities and differences between Latin American countries regarding the average intake of proteins from unprocessed and processed meats (Table 1). First of all, it was observed that, in all countries, the intake of unprocessed meats is significantly higher than processed meats, except for pork. Argentina and Brazil had the highest average levels of protein consumption from unprocessed beef (23.7 and 25.4 g/day, respectively) and processed beef (10.0 and 7.6 g/day, respectively). It should be noted that Argentina has the highest average consumption of processed pork (5.8 g/day).

Peru and Ecuador have the highest levels of protein intake from unprocessed poultry (27.5 and 24.2 g/day, respectively) and unprocessed fish (10.5 and 9.5 g/day, respectively), also, in these countries, the consumption of proteins derived from unprocessed pork (2.5 and 7.0 g/day, respectively) was higher than that of processed pork (1.5 and 3.7 g/day, respectively).

Venezuela and Colombia had similar average consumption in almost all the studied meats, except for the intake of unprocessed poultry, which in Venezuela (22.2 g/day) was significantly higher than in Colombia (19.2 g/day). Regarding the intake of unprocessed beef, a contrary behavior was observed, in which Colombia (17.0 g/day) had a significantly higher consumption than Venezuela (15.2 g/day).

Costa Rica and Chile, on average, had very low protein intake in almost all the meats under study. However, it should be noted that Costa Rica has the highest consumption of processed fish (3.9 g/day) and unprocessed pork (8.5 g/day) in the entire region.

### 3.7. Adequacy of Essential Amino Acid Intake

Essential amino acid intake in the overall ELANS population (Table 2) shows a low proportion of individuals with a deficient intake of specific essential amino acids, while the majority shows inadequate values by excess.

Using the PCA biplot method to analyze the intake of the nine essential amino acids by country (Figure 7). The Biplot is a graphic tool on which coordinates of variables and individuals obtained from a factor analysis are simultaneously presented. The axes of the graph are defined by the principal directions of the analysis. The variables are represented by vectors, and the angles between them are indicative of the strength and direction of the correlation between the variables. The projection of each individual on a vector reproduces approximately the value that the individual has in the corresponding variable.

The intakes of the different amino acids are narrow and directly correlated, i.e., all correlations > 0.9. The first factor ordered amino acid by intake and found a distinction between countries with high amino acid intake (Ecuador and Argentina, followed by Colombia, Venezuela, Brazil, and Peru) and low amino acid intake (Costa Rica and Chile). Within the second factor, Peru stands out by high consumption of tryptophan, which is in relation to the high consumption of poultry, fish, and cereals, whereas Brazil reports a higher intake of lysine and histidine, aligning with the high intake of beef and pork and low intake of cereals and eggs.

## 4. Discussion

This study’s results are relevant for understanding public health needs within the eight evaluated countries, particularly because protein and amino acid intake show a landscape on the preferences of the food sources of proteins, with a clear difference between the country’s individual access to animal and/or plant proteins. This fact needs to be taken into account when planning and designing regional or country’s public policies and programs.

Protein intake research results in an interesting field, especially in the light of emerging research that shows that diets with a certain essential amino acid profile, particularly those with a low intake of methionine, might be considered as part of the treatment for cancer patients, as high methionine intake might promote the growth of tumoral cells [26,27].

Likewise, considering the use of proteins of plant or animal origin requires the interpretation and an update about whether one is more environmentally friendly than the other and what the availability of each type exists within each ELANS country.

These aspects will require educating the population on the appropriate food combinations of animal and plant protein sources, the impact of those on the environment, and hopefully integrating with cultural appropriateness.

In this study, protein consumption was higher in men than women. Adolescents and young adults had the highest intake levels; middle-aged adults had decreased consumption as age increased; and older adults had strikingly lower protein consumption. Average protein intake increased with the ascent of the social scale. Our findings show lower values than those reported for the adult population worldwide by Miller et al. [28] in their article on animal food sources, based on the Global Dietary Database. These differences might be explained by the high variability between countries and their culinary traditions, the availability of foods or not, and the methodology used by Miller et al. [28] to address a follow-up average between 1990 and 2018. For instance, our data show dairy consumption between 7 and 14 g/day (depending on the country), whereas only yogurt accounts for 20 g/day globally, which is not surprising if we consider that the Middle East region and North African region are included in the average since those countries have a tradition for yogurt intake. Therefore, the total globe average might differ from values obtained in specific regions.

Proportional intake of animal- and plant-based proteins generally reflected agricultural, fishing, and ranching practices by country. A history in the Latin American region of cattle farm raising since its inception by the Spanish conquerors through countries such as Colombia and Venezuela and from there arriving in 1556 in Argentina allows an understanding of the tradition of beef eating in ELANS countries [29]. Since the very beginning, cattle adapted well in Latin America and went to form many of the culinary traditions that have been preserved to these days despite the many concerns of cattle farming and beef eating, including climate change, gas emissions, and the deleterious effects of excessive beef consumption. In addition to this, people consume according to what is available in the country. Therefore, the abundant consumption of beef, poultry, eggs, beans, rice, and many more also express what is available for the population and what they can access and buy [16].

No wonder animal proteins comprised nearly 70% of total daily protein intake in Argentina, Venezuela, Brazil, and Colombia, contrasting with <60% in Peru, Costa Rica, and Chile, at the moment of data collection. The average intake of animal proteins was generally higher from unprocessed sources than from processed sources. Pork was an exception; consumption of unprocessed versus processed meat did not differ significantly. These findings show the characteristics, quality, and sources of the most consumed proteins in ELANS countries, making it important to recognize the composition in terms of the amino acids delivered, as plant-based foods such as beans, grains, nuts, and soy are rich in some amino acids but may lack others [30].

Miller et al. [28] reported a wide variability within countries at the global level regarding animal sources proteins study. In 2018, the mean global consumption of red meat -unprocessed- was 51 g/day, with a 16-fold variation across several geographical regions. From this finding, we learned that Latin America and the Caribbean had an increased intake of 1.29 servings per week of unprocessed red meat, whereas the sub-Saharan region had an increase of 0.06 servings per week, and Southeast Asia and East Asia increased the intake of 4.12 weekly servings [28]. Therefore, it is no surprise the huge variability in amino acid intake across countries due to the different sources of protein intake as well as the amount of these sources across countries.

Different components of an animal-based diet—red meat, fish, eggs, and dairy products—also have different properties and effects on health. For example, the risk for ischemic heart disease was positively associated with the consumption of red meat but inversely associated with the consumption of yogurt, cheese, and eggs [31]. In terms of sustainability as a food source, animal protein-based diets require large areas of dedicated land, water, and fossil energy for production and transportation, in turn releasing large amounts of potentially harmful greenhouse gases [12]. However, new research also highlights the same potential threat from extensive mono-agriculture [19].

It is important to highlight that essential amino acids cannot be synthesized by the body, so they must be provided through protein-containing foods. The WHO has established adult requirements for daily intake of the nine essential amino acids [3]. Animal-sourced dietary proteins provide all essential amino acids. Some plant-based protein sources contain low levels of certain essential amino acids (low lysine in rice and other grains or low methionine in beans), so dietary adequacy of essential amino acids depends on eating complementary protein sources [32]. Unprocessed foods are widely considered healthier than processed foods. Unprocessed foods are whole foods consumed in their natural state, although some may be minimally altered by the removal of inedible parts or by drying, cooking, freezing, or pasteurizing them for safe storage before consumption. Fresh or frozen fruits and vegetables, raw chicken, fish, and whole cuts of red meats, eggs, and nuts are examples of unprocessed and minimally processed foods. Processed foods are changed from their natural state by adding sugar, salt, oil, or substances such as preservatives. Notably, adult diets in the United States have recently been described to contain nearly 60% processed foods [33]. Consumption of highly-processed foods has been associated with an increased risk of cardiovascular disease, all-cause mortality, and colorectal cancer [34,35]. Again, public health education and policies in Latin America can be modified to promote the intake of high-quality proteins through diets that are friendly to the planet.

As challenging as defining a healthy diet is, and the dilemma of plant-based vs. animal-based protein consumption, in the realm of the ongoing research regarding the benefits of plant-based diets, but also the challenges of extensive agriculture and farming (cattle and fishing) as the big promotors of climate changes [17], a new parameter might be introduced into the prescription of healthy diets for preserving the planet’s health and could also be taken into account for the design of public policies, food and nutrition guidelines and regulatory issues. We refer to education to the population on these issues so people make smart choices and avoid food waste on a regular basis.

Our study is a unique one within the Latin American region and provides nationally representative samples for each country, the obtainment of two non-consecutive 24 h recalls and the multiple pass methodology gives this study the strength to examine the associations between different sources of protein and other factors. In addition, the unified methodology allows the analysis of countries to gain a regional perspective. In addition, it constitutes a great baseline data source for pre-pandemic analysis that allows us to know about the intake of foods, particularly of protein, within ELANS countries. This should be part of the regional assessment that must occur in the future, as Latin America has a diverse population that needs to be addressed when designing regional policies. It also has some limitations, such as being a cross-sectional study, so we cannot investigate causalities but rather associations that can be explored in future research.

## 5. Conclusions

Although protein consumption was generally adequate for adolescents and adults in the ELANS countries of Latin America, there are still some challenges in terms of quality, variety, and quantities. For challenged groups within the population, particularly older adults, public health policies and system changes are essential to ensure dietary adequacy. In addition, the impact of protein production on the environment should be considered in the future design of public policies and programs. Public health education and guidelines can provide direction for healthy and adequate protein intake in ways that are friendly to the environment and alleviate climate change.

## Figures and Tables

**Figure 1 nutrients-15-00669-f001:**
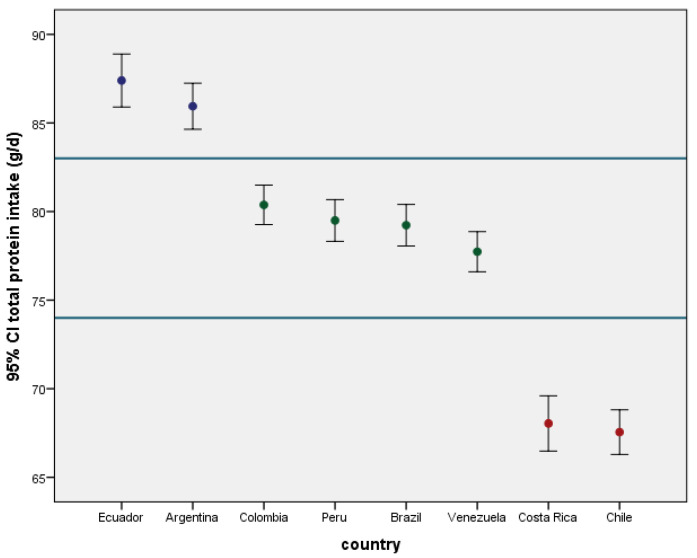
Mean protein intake (g/day) by ELANS countries with standard error bar.

**Figure 2 nutrients-15-00669-f002:**
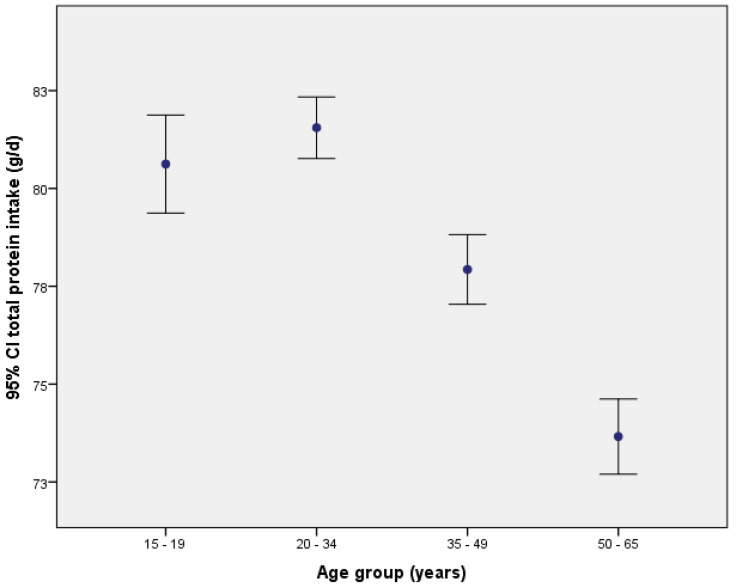
Mean protein intake by age group.

**Figure 3 nutrients-15-00669-f003:**
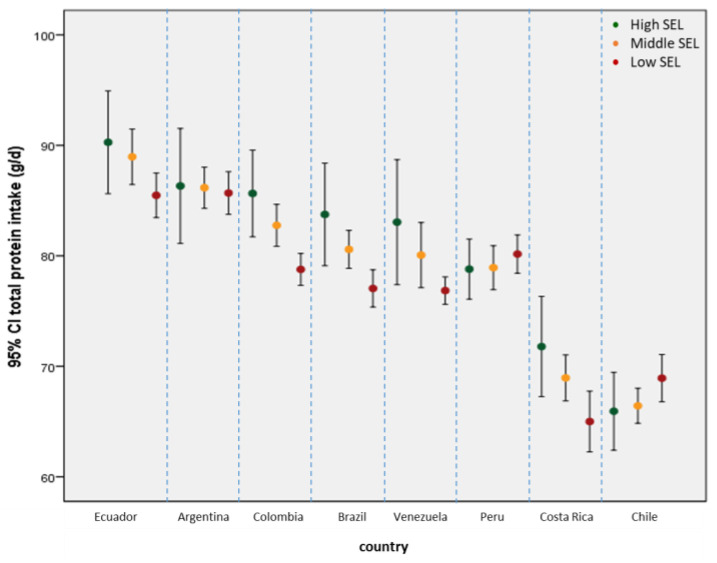
Mean daily protein intake (g/day) by socio-economic level by ELANS countries.

**Figure 4 nutrients-15-00669-f004:**
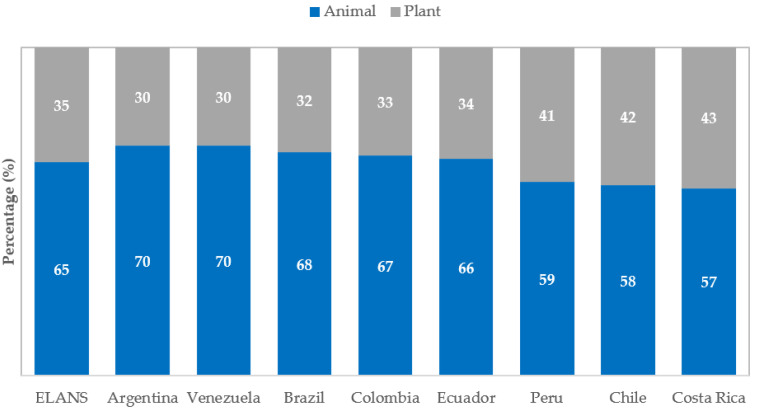
Proportionate intake of plant- and animal-based protein (as percentages) by ELANS countries.

**Figure 5 nutrients-15-00669-f005:**
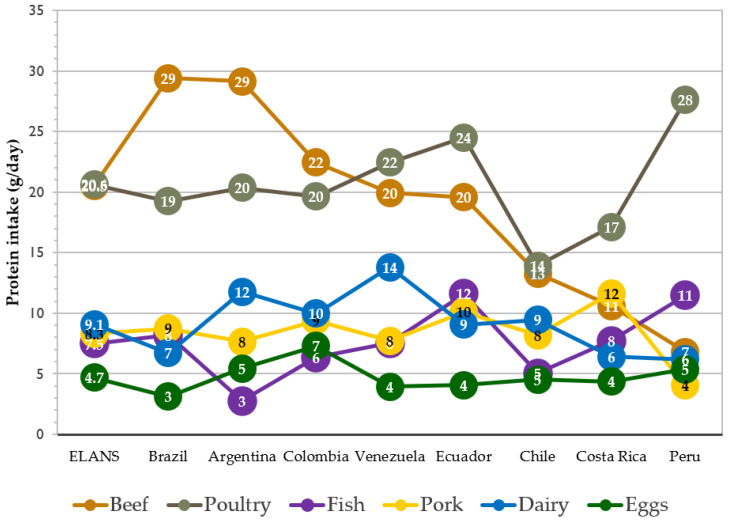
Mean daily animal protein intake (g/day) by food group and ELANS countries.

**Figure 6 nutrients-15-00669-f006:**
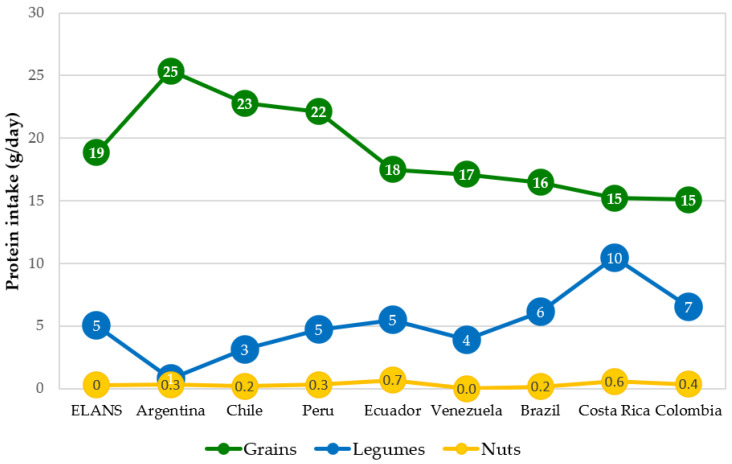
Mean daily plant protein intake (g/day) by food group and ELANS countries.

**Figure 7 nutrients-15-00669-f007:**
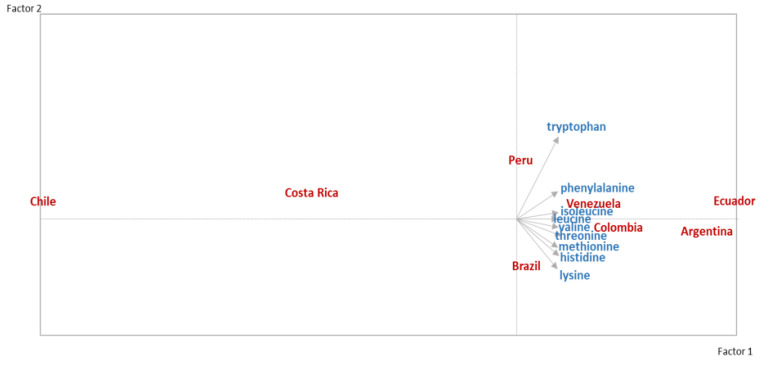
PCA Factorial Biplot: essential amino acid intake.

**Table 1 nutrients-15-00669-t001:** Mean daily protein intake (g/day) of processed and unprocessed meats by ELANS countries.

Country	Beef		Poultry		Fish		Pork	
	U	P	U	P	U	P	U	P
ELANS	16.8	6.0	20.4	0.2	6.0	1.6	4.3	4.6
	(0.19)	(0.11)	(0.19)	(0.02)	(0.15)	(0.06)	(0.12)	(0.06)
Argentina	23.7	10.0	20.3	0,27	2.1	0.9	2.2	5.8
	(0.59)	(0.33)	(0.60)	(0.07)	(0.30)	(0.14)	(0.20)	(0.18)
Brazil	25.4	7.6	19.1	0.22	7.6	0.0	4.0	5.0
	(0.51)	(0.27)	(0.43)	(0.05)	(0.45)	-	(0.28)	(0.15)
Chile	9.4	5.2	13.9	0.0	3.5	1.8	3.2	5.7
	(0.48)	(0.28)	(0.59)	-	(0.34)	(0.19)	(0.33)	(0.20)
Colombia	17.0	8.1	19.2	0.0	4.8	1.7	4.9	5.3
	(0.43)	(0.39)	(0.47)	-	(0.41)	(0.18)	(0.32)	(0.17)
Costa Rica	6.8	4.5	16.6	0.83	4.1	3.9	8.5	4.3
	(0.36)	(0.26)	(0.55)	(0.17)	(0.34)	(0.30)	(0.51)	(0.15)
Ecuador	18.9	1.2	24.2	0.6	9.5	2.7	7.0	3.7
	(0.56)	(0.19)	(0.52)	(0.16)	(0.43)	(0.25)	(0.51)	(0.23)
Peru	6.3	0.0	27.5	0.0	10.5	1.4	2.5	1.5
	(0.26)	-	(0.40)	-	(0.44)	(0.15)	(0.25)	(0.08)
Venezuela	15.2	7.6	22.2	0.0	5.4	2.5	3.8	4.7
	(0.47)	(0.31)	(0.50)	-	(0.44)	(0.23)	(0.37)	(0.13)

U: Unprocessed; P: Processed, (Standard Error of Mean in parenthesis).

**Table 2 nutrients-15-00669-t002:** Adequacy of intake for essential amino acids in ELANS.

Essential Amino Acid	WHO Requirement (mg/kg/day)	Percentage of the Sample		
		<90 (Deficient)	90–110 (Normal)	>110 (Excessive)
Leucine	39	5.6	5.3	89.1
Lysine	30	4.9	4.2	90.8
Valine	26	5.9	5.7	88.4
Phenylalanine	25	8.0	7.5	84.5
Isoleucine	20	3.9	3.8	92.3
Threonine	15	2.5	2.5	95.0
Histidine	10	1.9	2.1	96.0
Methionine	10	4.3	3.8	91.9
Tryptophan	4	1.9	2.1	96.1

## Data Availability

Data presented in this study are available upon request from the corresponding author. Data are not publicly available due to privacy or ethical restrictions.

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
