# Peer review of "Contribution of Proteins to the Latin American Diet: Results of the ELANS Study"

_nutrients, 2023, doi:10.3390/nu15030669_

Round 1
Reviewer 1 Report
The topic is interesting, however, to improve the quality of the manuscript, there are several suggestions to be addressed:
The English should be checked throughout - for example, the first paragraph in Discussion
All numbers in text, figures, tables should be corrected! As this is an English journal, decimal points should be used instead of commas (e.g., line 63 - 0.8 instead of 0,8)
Line 186: Legend should go under figure
Lines 197-201: Do you have another figure here? The text does not match the existing Figure 3
Line 216: if in descending order, as above (lines 214-215), than change to “Peru, Chile, and Costa Rica”. The same in Abstract
Based on Figure 5:
Line 231: “... in beef and poultry (11 to 17 g/day)” instead of “... in meat and poultry (13 to 17 g/day)”
Line 232: “... only 7 g/day of beef” instead of “... only 7 g/day of meats”
Lines 236, 256: numbers should be placed on vertical lines in Figure 5 and Figure 6
Lines 248-255: many inaccuracies as the text does not match the numbers in Table 1! “respectively” should be used...
For example, the first sentence (lines 248-250) should be “Ecuador, Argentina, and Brazil had the highest mean protein consumption from unprocessed beef (18.9, 23.7, and 25.4 g/day, respectively) ...” instead of “Brazil, Colombia, and Argentina had the highest mean protein consumption from unprocessed meats (22,0 to 23,2 g/day) ...”
Lines 261-262: same as above - it should be “Ecuador and Costa Rica showed the highest consumption of processed fish (2.7 and 3.9 g/day, respectively)” instead of “Costa Rica and Chile showed the highest consumption of processed fish (2,4 to 3,1 g/day)”
In Discussion you say:
Lines 326-328: “However, plant-based proteins from foods like beans, grains, nuts, and soy are rich in some amino acids but may lack others.”
Lines 332-335: Some plant-based protein sources contain low levels of certain essential amino acids (low lysine in rice and other grains or low methionine in beans), so dietary adequacy of essential amino acids depends on eating complementary protein sources.[26]” In addition, nuts are deficient in lysine and methionine (doi: 10.1039/c7fo01967j).
These are the reasons you should include in Discussion (in Abstract and Conclusions) the following scientific findings:
The essential amino acid, methionine, is important for cancer cell growth and metabolism... methionine restriction inhibits cancer cell growth and may enhance the efficacy of chemotherapeutic agents (doi: 10.3390/nu12030684).
In humans, vegan diets, which can be low in methionine, may prove to be a useful nutritional strategy in cancer growth control (doi: 10.1016/j.ctrv.2012.01.004).
Among breast cancer survivors, decreased methionine intake after breast cancer diagnosis was associated with lower risk of all-cause and breast cancer mortality (doi: 10.3390/nu14224747).
Methionine-restricted diet substantially diminished metastatic potential and holds substantial promise for cancer treatment (doi: 10.1093/carcin/bgy085).
An emerging consensus of animal and human data suggests that, contrary to long-held popular beliefs, lower protein consumption is more beneficial for health and longevity than high protein consumption... It is now becoming clear that dietary protein quality – the specific amino acid composition of the dietary protein – has a profound effect on metabolic health and longevity in mammals... restriction of methionine, branched chain amino acids (BCAAs) or tryptophan can improve healthspan and lifespan in rodents. While only a few human studies on these types of diets have been conducted, preliminary evidence suggests that restriction of methionine or BCAAs may also have metabolic benefits in humans (doi: 10.1016/j.tma.2021.05.001).
Line 373: “obtention”?
Author Response
The authors would like to thank the reviewers for their precious time and invaluable comments. We have carefully addressed all the comments. The corresponding changes and refinements were made in the revised paper. They are also summarized in our response below.
The topic is interesting, however, to improve the quality of the manuscript, there are several suggestions to be addressed:
The English should be checked throughout - for example, the first paragraph in Discussion
All numbers in text, figures, tables should be corrected! As this is an English journal, decimal points should be used instead of commas (e.g., line 63 - 0.8 instead of 0,8).
R: We thank the reviewer for pointing this out. We have made the change.
Line 186: Legend should go under figure
R: We have made the change in all the figures.
Lines 197-201: Do you have another figure here? The text does not match the existing Figure 3
R: This observation is correct. We clarified in the text and matched the figure with the explanation.
Line 216: if in descending order, as above (lines 214-215), than change to “Peru, Chile, and Costa Rica”. The same in Abstract
R: We have made the change both in Results and Abstract.
Based on Figure 5:
Line 231: “... in beef and poultry (11 to 17 g/day)” instead of “... in meat and poultry (13 to 17 g/day)”
R: We have made the change.
Line 232: “... only 7 g/day of beef” instead of “... only 7 g/day of meats”
R: We have made the change.
Lines 236, 256: numbers should be placed on vertical lines in Figure 5 and Figure 6
R: We have added the numbers on vertical lines.
Lines 248-255: many inaccuracies as the text does not match the numbers in Table 1! “respectively” should be used...
For example, the first sentence (lines 248-250) should be “Ecuador, Argentina, and Brazil had the highest mean protein consumption from unprocessed beef (18.9, 23.7, and 25.4 g/day, respectively) ...” instead of “Brazil, Colombia, and Argentina had the highest mean protein consumption from unprocessed meats (22,0 to 23,2 g/day) ...”
R: We have modified the table 1 adding the standard error value and we have changed all the text about it. We hope that it is now clearer than before.
Lines 261-262: same as above - it should be “Ecuador and Costa Rica showed the highest consumption of processed fish (2.7 and 3.9 g/day, respectively)” instead of “Costa Rica and Chile showed the highest consumption of processed fish (2,4 to 3,1 g/day)”
R: We have rewritten the text.
In Discussion you say:
Lines 326-328: “However, plant-based proteins from foods like beans, grains, nuts, and soy are rich in some amino acids but may lack others.”
Lines 332-335: Some plant-based protein sources contain low levels of certain essential amino acids (low lysine in rice and other grains or low methionine in beans), so dietary adequacy of essential amino acids depends on eating complementary protein sources.[26]” In addition, nuts are deficient in lysine and methionine (doi: 10.1039/c7fo01967j).
These are the reasons you should include in Discussion (in Abstract and Conclusions) the following scientific findings:
The essential amino acid, methionine, is important for cancer cell growth and metabolism... methionine restriction inhibits cancer cell growth and may enhance the efficacy of chemotherapeutic agents (doi: 10.3390/nu12030684).
In humans, vegan diets, which can be low in methionine, may prove to be a useful nutritional strategy in cancer growth control (doi: 10.1016/j.ctrv.2012.01.004).
Among breast cancer survivors, decreased methionine intake after breast cancer diagnosis was associated with lower risk of all-cause and breast cancer mortality (doi: 10.3390/nu14224747).
Methionine-restricted diet substantially diminished metastatic potential and holds substantial promise for cancer treatment (doi: 10.1093/carcin/bgy085).
An emerging consensus of animal and human data suggests that, contrary to long-held popular beliefs, lower protein consumption is more beneficial for health and longevity than high protein consumption... It is now becoming clear that dietary protein quality – the specific amino acid composition of the dietary protein – has a profound effect on metabolic health and longevity in mammals... restriction of methionine, branched chain amino acids (BCAAs) or tryptophan can improve healthspan and lifespan in rodents. While only a few human studies on these types of diets have been conducted, preliminary evidence suggests that restriction of methionine or BCAAs may also have metabolic benefits in humans (doi: 10.1016/j.tma.2021.05.001).
R: We appreciate the reviewer’s insightful suggestion and agree that it would be useful to enrich the discussion section. This is a very interesting suggestion and we discussed it a lot within our team. We feel that for the current paper this would go too far but we have included two references in the discussion section and once more, are thankful for having received this valuable idea from the reviewer.
Line 373: “obtention”?
R: We have modified the word “obtention” for “obtainment”.
Sincerely yours,
On behalf of the ELANS Study team
Marianella Herrera Cuenca
Associate Professor Universidad Central de Venezuela, Caracas, Venezuela
Visiting Lecturer Framingham State University, Framingham, MA, USA
Adjunct Professor Simmons University, Boston, MA, USA
Reviewer 2 Report
Comment: Thank you for the opportunity to review the manuscript entitled “Contribution of proteins to the Latin American diet: Result of the ELANS study” submitted for publication to Nutrients. The research sought to characterize the dietary sources of protein in eight Latin American countries (i.e., Argentina, Brazil, Chile, Colombia, Costa Rica, Ecuador, Peru, Venezuela) through survey data. 9218 participants have been recruited during the research. The first purpose of the research is to quantify per person daily protein consumption by country, and the second purpose is to quantify proportional intake of protein by source, amount and processing, as well as to determine adequacy of protein/essential amino acid intake. The purpose of the research is interesting. However, the manuscript does not provide essential information related to data collection and the methods applied. Further, data have been collected from 2014 to 2015, which makes results quite old and not useful in the current context. Results are simple and not critically evaluated/interpreted. Please, consider the subsequent point-by-point comments. I have found several major issues and criticalities in the current research.
Introduction: Please, consider the Instruction for Authors provided by MDPI. For instance, references should be included before the full stop (e.g., [1].) and not after the full stop (e.g., .[1]). Also, numbers should be divided with full stops (e.g., 0.8 g/kg) and not with commas (e.g., 0,8 g/k).
L. 75-82. It is not clear what the authors are willing to say. They start the paragraph by saying “without going in depth…”. But it is not clear what the issue is about: are plant-based proteins more environmentally friendly compared to animal-based proteins? Could you please introduce such a topic in a clear and comprehensive manner, also by providing examples from previous literature and providing examples related to environmental impacts of proteins production compared to similar functional units?
L. 82. It is not clear the nexus between “climate change”, “geopolitical conflicts” and “protein production”. Please, develop further such an interesting concept.
The purpose of the research is provided in a comprehensive manner in L. 83-91. However, the “Introduction” does not provide sufficient background information and does not include all relevant references. A proper section devoted to the “Literature review” or the “Theoretical background” is missing. Further, the authors must highlight the novelty/originality of such a research, and the audience for the research. I wonder: why readers should be interested in such a topic? Will the authors discuss public authorities’ implications, as well as environmental implications?
Materials and Methods.
L. 94-100. The authors refer to the previous article conducted in the field of the ELANS research, and I agree. However, although such a reference is complete and interesting (i.e., [19]), I would add some more details related to the research area of the research, as well as the “study protocol, etc.”. Few but essential information to strengthen the research.
Also, data from 2014 to 2015 are somehow old, quite ten years ago. How could you cope with such an issue? Could you consider that eating consumption behavior in the selected countries have not been modified in the last ten years? How can I use such data (2014-2015) in the 2023 context? Are results useful and comparable? How the context described in [19] has changed from 2016 to 2023?
L. 106-108. The authors refer to the questionnaire provided in [19] for demographics data. However, how data related to proteins have been collected? The aims of the research is: “(1) to quantify per-person daily protein consumption by country and by sociodemographic factors and to establish between countries variability; (2) to quantify proportional intake of proteins by food groupings, protein source, and processing; and (3) to determine adequacy of protein/essential amino acid intake.” Hence, how have you collected such information? I cannot find any details related to the questions asked to respondents, to the typology of possible answers, etc. It seems that an essential aspect of the methodology is missing, meaning the variables asked to respondents to conduct the research.
Results: Results have been analyzed in a quite descriptive manner, which makes reading simple but rather basic. I cannot understand how results have been calculated, considering that there is not information related to the questionnaire asked to respondents. Further, Figure (i.e., 3, 4, 5, 6) are quite simple for a scientific article. Is seems rather a report and not a scientific article.
How have you depicted the details related to the essential amino acids from the respondents’ answers? How have you assumed the amino acid content embedded within each food category? It is not clear and should be developed further, both in the section “Materials and Methods” and in the section “Results” (sub-section 3.7.).
Figure 7 must be better described in the section “Results”.
Discussion: “Discussion” appear like a summary of previous results. Readers cannot find any managerial, public authorities or theoretical implications. How the collected data could be used? Which was the aim to collect such quantitative data? Could you provide some possible solutions or improvements in proteins consumption behaviors among countries?
Supplementary Materials: The link “www.mdpi.com/xxx/s1” (L. 385) does not work.
Author Response
The authors would like to thank the reviewers for their precious time and invaluable comments. We have carefully addressed all the comments. The corresponding changes and refinements were made in the revised paper. They are also summarized in our response below.
Introduction: Please, consider the Instruction for Authors provided by MDPI. For instance, references should be included before the full stop (e.g., [1].) and not after the full stop (e.g., .[1]). Also, numbers should be divided with full stops (e.g., 0.8 g/kg) and not with commas (e.g., 0,8 g/k).
R: We thank the reviewer for pointing this out. We have made the change.
L. 75-82. It is not clear what the authors are willing to say. They start the paragraph by saying “without going in depth…”. But it is not clear what the issue is about: are plant-based proteins more environmentally friendly compared to animal-based proteins? Could you please introduce such a topic in a clear and comprehensive manner, also by providing examples from previous literature and providing examples related to environmental impacts of proteins production compared to similar functional units?
R: We realized that we have not expressed ourselves clearly enough in our text. To avoid any misunderstandings and to address the reviewer’s concern, we clarified the existing challenge between animal sources of protein greenhouse emissions and the extensive mono-agriculture being also a source of the same.
L. 82. It is not clear the nexus between “climate change”, “geopolitical conflicts” and “protein production”. Please, develop further such an interesting concept.
R: We agree and have added a brief explanation about these topics. We reinforced the point, highlighting the nexus between war, interruption of supply chains and increased of prices within the global food markets
The purpose of the research is provided in a comprehensive manner in L. 83-91. However, the “Introduction” does not provide sufficient background information and does not include all relevant references. A proper section devoted to the “Literature review” or the “Theoretical background” is missing. Further, the authors must highlight the novelty/originality of such a research, and the audience for the research. I wonder: why readers should be interested in such a topic? Will the authors discuss public authorities’ implications, as well as environmental implications?
R: We reinforced the introduction, by reinforcing our literature review, however for an original research article a “theoretical background” within the introduction would be too long and going away from the instructions for authors.
We highlighted within the introduction there are few studies on protein intake in Latin America, particularly with a standardized unified methodology. In addition, we explained the relevance acquired by this study after the COVID-19 pandemic, as it is a valuable baseline data that might be useful in planning and design policies while comparing with future studies (when available). Also, we explored how pertinent these results are in the discussion where we also mention the environmental implication of both types of protein consumption: vegetal and animal.
Materials and Methods.
L. 94-100. The authors refer to the previous article conducted in the field of the ELANS research, and I agree. However, although such a reference is complete and interesting (i.e., [19]), I would add some more details related to the research area of the research, as well as the “study protocol, etc.”. Few but essential information to strengthen the research.
R: We have made the change, and added details about the protocol in the Methodology section.
Also, data from 2014 to 2015 are somehow old, quite ten years ago. How could you cope with such an issue? Could you consider that eating consumption behavior in the selected countries have not been modified in the last ten years? How can I use such data (2014-2015) in the 2023 context? Are results useful and comparable? How the context described in [19] has changed from 2016 to 2023?
R: The data from ELANS is 8-9 years old, however until the best of our knowledge there are few studies that integrate regional data in Latin America. In addition, this data is now very valuable since it constitutes a pre-pandemic regional baseline to evaluate and compare the post-pandemic status of the macro and micronutrient intake. In ELANS countries, there are no new data collection, again until the best of our knowledge, unifying methodologies to analyze the regional impact.
L. 106-108. The authors refer to the questionnaire provided in [19] for demographics data. However, how data related to proteins have been collected? The aims of the research is: “(1) to quantify per-person daily protein consumption by country and by sociodemographic factors and to establish between countries variability; (2) to quantify proportional intake of proteins by food groupings, protein source, and processing; and (3) to determine adequacy of protein/essential amino acid intake.” Hence, how have you collected such information? I cannot find any details related to the questions asked to respondents, to the typology of possible answers, etc. It seems that an essential aspect of the methodology is missing, meaning the variables asked to respondents to conduct the research.
R: Thank you for this question, we reinforced in the text, but if the reviewer is interested in know the full extent of the 24 Hours recall and Multiple Pass Methodology, we recommend to read the references 20 and 21. Nevertheless, a 24 HRS recall assess the intake of the previous 24 hours, and the trained interviewers get detailed information on all food and beverages, preparations and supplements consumed. To diminish the memory errors, 2 consecutive 24 hour recalls were performed, and the multiple pass method guaranteed that no step of the recall was forgotten.
Results: Results have been analyzed in a quite descriptive manner, which makes reading simple but rather basic. I cannot understand how results have been calculated, considering that there is not information related to the questionnaire asked to respondents. Further, Figure (i.e., 3, 4, 5, 6) are quite simple for a scientific article. Is seems rather a report and not a scientific article.
R: Again, as in the previous answer, we reinforced the fact that a 24 Hour recall we can obtain the general intake of foods, however, we must remember that from this recall we will identify which proportion of foods corresponds to proteins, carbohydrates, fats, and micronutrients. Eg: What did you eat during the last 24 hours ? The respondent answer: I ate just at lunch, 12 noon, one cup of rice, 60 grams of beef and ½ cup of broccoli. I f that is the answer we know that yesterday, the respondent ate 60 grams of beef, which will be analyzed in terms of the composition (with the software NDSR was analyzed into its components: proteins, fats, etc, and also amino acid present)
How have you depicted the details related to the essential amino acids from the respondents’ answers? How have you assumed the amino acid content embedded within each food category? It is not clear and should be developed further, both in the section “Materials and Methods” and in the section “Results” (sub-section 3.7.).
R: This was clarified in the text and previous answers
Figure 7 must be better described in the section “Results”.
R: We reinforced in the text.
Discussion: “Discussion” appear like a summary of previous results. Readers cannot find any managerial, public authorities or theoretical implications. How the collected data could be used? Which was the aim to collect such quantitative data? Could you provide some possible solutions or improvements in proteins consumption behaviors among countries?
R: Thank you for this question, we reinforced the relevance of using this data as a pre-pandemic baseline, and to establish future comparison, when new data is available. This data can be used to address with equity programs and policies to improve protein consumption in ELANS countries.
Supplementary Materials: The link “www.mdpi.com/xxx/s1” (L. 385) does not work.
R: We need to wait for MDPI to activate that link, the authors have no power to activate that link
Sincerely yours,
On behalf of the ELANS Study team
Marianella Herrera Cuenca
Associate Professor Universidad Central de Venezuela, Caracas, Venezuela
Visiting Lecturer Framingham State University, Framingham, MA, USA
Adjunct Professor Simmons University, Boston, MA, USA
Round 2
Reviewer 1 Report
The authors addressed the suggestions and the manuscript is much improved. Please correct the following mistakes:
Line 162: Parentheses missing for references
Line 263: ‘and’ instead of ‘y’
Line 287, Figure 6: to eliminate confusion, ‘grains’ should be used as ‘cereals’ are breakfast foods
Line 296 (Figure 6): ‘Legumes’
Line 381: ‘our data show’
Line 389: ‘raising’ instead of ‘rising’
Author Response
The authors addressed the suggestions and the manuscript is much improved. Please correct the following mistakes:
Line 162: Parentheses missing for references
R: The references were duplicated. We have made the change.
Line 263: ‘and’ instead of ‘y’
R: We have made the change.
Line 287, Figure 6: to eliminate confusion, ‘grains’ should be used as ‘cereals’ are breakfast foods
Line 296 (Figure 6): ‘Legumes’
R: We have made the changes in figure 6.
Line 381: ‘our data show’
R: We have made the change.
Line 389: ‘raising’ instead of ‘rising’
R: We have made the change.